# Functional Cardiovascular Characterization of the Common Marmoset (*Callithrix jacchus*)

**DOI:** 10.3390/biology12081123

**Published:** 2023-08-11

**Authors:** Lina Klösener, Sabine Samolovac, Ina Barnekow, Jessica König, Amir Moussavi, Susann Boretius, Dieter Fuchs, Astrid Haegens, Rabea Hinkel, Matthias Mietsch

**Affiliations:** 1Laboratory Animal Science Unit, German Primate Center, Leibniz Institute for Primate Research, 37077 Göttingen, Germanymmietsch@dpz.eu (M.M.); 2DZHK (German Center for Cardiovascular Research), Partner Site Göttingen, 37075 Göttingen, Germany; 3Institute for Animal Hygiene, Animal Welfare and Farm Animal Behavior, University of Veterinary Medicine, 30173 Hannover, Germany; 4Functional Imaging Laboratory, German Primate Center, Leibniz Institute for Primate Research, 37077 Göttingen, Germany; 5Johann-Friedrich-Blumenbach Institute of Zoology and Anthropology, Georg August University, 37077 Göttingen, Germany; 6FUJIFILM VisualSonics Inc., 1114 AB Amsterdam, The Netherlands; 7Transonic Inc., Ithaca, NY 14850, USA

**Keywords:** hemodynamic measurement, common marmoset, *Callithrix jacchus*, magnetic resonance imaging, echocardiography, animal model, cardiac, pressure–volume loops, heart failure

## Abstract

**Simple Summary:**

Animal models reflecting the human situation as accurately as possible are still essential to develop new therapies for diseases such as heart failure. The common marmoset (*Callithrix jacchus*), a commonly used small non-human primate, resembles humans in a variety of aspects. However, its cardiovascular system has not yet been sufficiently studied. It is imperative, though, to have a full understanding of this animal’s cardiovascular system and to have tools to accurately survey it. Therefore, for the first time, cardiac catheterizations along with intracardiac pressure–volume loop (PV loop) measurements were performed that provide real-time heart function data. By developing a protocol for cardiac catheterization (PV loop) in this animal species, the cardiac function in young, healthy animals was assessed. In the same animals, the obtained results were compared with generally established methods, namely, magnetic resonance imaging (MRI) and echocardiography. All three methods were suitable to describe cardiac function in the common marmoset; however, a comparison of the results revealed certain differences between the three techniques that are important for future research. Depending on the study objective, each method has its advantages and disadvantages and should, therefore, be carefully selected. The results provide the basis to establish the common marmoset as a promising primate model for cardiovascular disease.

**Abstract:**

Appropriate cardiovascular animal models are urgently needed to investigate genetic, molecular, and therapeutic approaches, yet the translation of results from the currently used species is difficult due to their genetic distance as well as their anatomical or physiological differences. Animal species that are closer to the human situation might help to bridge this translational gap. The common marmoset (*Callithrix jacchus*) is an interesting candidate to investigate certain heart diseases and cardiovascular comorbidities, yet a basic functional characterization of its hemodynamic system is still missing. Therefore, cardiac functional analyses were performed by utilizing the invasive intracardiac pressure–volume loops (PV loop) system in seven animals, magnetic resonance imaging (MRI) in six animals, and echocardiography in five young adult male common marmosets. For a direct comparison between the three methods, only data from animals for which all three datasets could be acquired were selected. All three modalities were suitable for characterizing cardiac function, though with some systemic variations. In addition, vena cava occlusions were performed to investigate the load-independent parameters collected with the PV loop system, which allowed for a deeper analysis of the cardiac function and for a more sensitive detection of the alterations in a disease state, such as heart failure or certain cardiovascular comorbidities.

## 1. Introduction

Cardiovascular diseases are the number one cause of death worldwide, with increasing incidence especially in the Western world [1]. Hypertension, coronary heart disease, myocardial infarction, and heart failure pose life-threatening conditions for millions of people [2]. Additionally, due to the worldwide extending life span of the elderly, the appearance of age-associated cardiovascular diseases is expected to significantly rise during the next decades [3]. The health, economic, and social consequences of these diseases constitute significant challenges for countries across the globe. To provide the optimal treatment and prevention of severe cardiac diseases, not only investigations of genetic causes, molecular pathways, and therapeutic targets but also the testing of new treatment options are urgently needed.

The commonly used animal models for these research questions are typically rodent or porcine species. The possibility to generate genetically modified mice enables the basic clarification of genetic and molecular pathways, whereas—due to their size—larger animal models are often used to test new therapeutic interventional or surgical approaches [4,5,6]. However, the translation of research results is often complicated due to the genetic distance or the anatomical and physiological differences of these animal species compared to the human situation.

Due to their genetic proximity to humans, comparable anatomic features, and often-similar disease phenotypes, non-human primates (NHPs) are considered to mirror the human situation most closely and would—in theory—constitute the ideal animal model. The common marmoset (*Callithrix jacchus*) is a small (300–500 g bodyweight), short-lived (for a life span comparison, see Table 1 and Figure 1) NHP species that is currently the most commonly used. It displays a close genetic proximity to humans compared to more commonly used rodent species and has a low zoonotic potential [7,8]. This animal species is already established in a variety of research areas like inter alia neurological research, behavioral research, and stem cell research [9,10,11,12,13,14,15], and the recently discovered possibility to generate transgenic animals broadened its potential use even further [16].

In contrast to larger, longer-lived primate species, the common marmoset exhibits a fast gestational period and, thereby, might be advantageous to other NHP models. Additionally, the common marmoset might be a potential animal model in gerontology and metabolic research, as it displays various aging symptoms beginning as early as the age of 6–8 years [11], including immunosenescence [17] and blood pressure changes [18] with advancing age or increased bodyweight, thereby reflecting the human situation.

However, investigations into its cardiovascular system are comparably scarce. Recent studies revealed similarities of common marmosets to humans regarding heart anatomy and topography; e.g., similar relative heart weight and ECG patterns, branching of the large vessels, heart anatomy and—on a cellular level—collagen quantity and distribution [19,20,21]. Furthermore, the first report about cardiovascular magnetic resonance imaging (MRI) by Moussavi et al. suggests striking similarities to the human situation in terms of heart senescence [22]. These findings could predispose this animal species for use in cardiac and cardiovascular research. Therefore, basic characterizations of this NHP’s cardiovascular system are necessary.

Invasive cardiac function measurements, so-called pressure–volume loops (PV loops), are a well-established and common technique for the comprehensive assessment of cardiovascular function in different animal models [23,24,25,26]. Although being more invasive, in contrast to commonly used cardiac diagnostic methods like MRI and echocardiography, they offer the opportunity of a small, portable real-time hemodynamic setup for investigating cardiac function including the measurement of preload-independent parameters, e.g., the end-systolic pressure–volume relationship (ESPVR) or preload recruitable stroke work (PRSW). Due to these advantages, it is usually considered as advantageous to assess comprehensive cardiac function in different research settings compared to the non-invasive imaging modalities, MRI and echocardiography, which are largely load-dependent [27,28].

Therefore, the aim of this study was to establish PV loop measurements in the common marmoset primates with potential application in future cardiac studies. In order to validate and classify the collected data, MRI and cardiac echography of the same animals were performed, and the results from the different methods were compared. To the best of the authors’ knowledge, admittance-based PV loop measurements have not been conducted in marmosets before. The provided surgery protocol as well as the reported results characterize the hemodynamic conditions of this interesting animal model and provide guidance and basic data, especially with regard to future cardiovascular investigations. The obtained outcomes contribute to the consolidation of the common marmoset in these fields of research.

**Table 1 biology-12-01123-t001:** Life span of humans and different animal models in comparison. Marmoset = *Callithrix jacchus*, pig = *Sus scrofa domesticus*, mouse = *Mus musculus*, w = weeks, m = months, y = years.

Species	Infant	Juvenile	Adolescent	Adult	Aged
Human [29,30]	<12 m	1–12 y	12–18 y	18–65 y	>65 y
Marmoset [31]	<3 m	3–5 m	0.5–2 y	2–8 y	>8 y
Pig [32]	<2.5 m	2.5–4.5 m	4.5–6.5 m	>6.5 m	
Mouse [33]	<3 w	3–4 w	1–3 m	3–20 m	>20 m

## 2. Materials and Methods

### 2.1. Ethics

Animals were fostered and kept at the German Primate Center, Göttingen, Germany. Husbandry and experiments were conducted in compliance with Directive 2021/63/EU of the European Union and the German Animal Welfare Act and, therefore, met the regulations of the European Animal Research Association. The animal study protocol was approved by the institutional animal welfare committee and, subsequently, by the Lower Saxony State Office for Consumer Protection and Food Safety (LAVES; reference number 33.19-42502-04-20/3458). Experimental procedures were in accordance with regulations of the German Legislature.

### 2.2. Animals, Husbandry and Housing

Marmosets were conventionally housed in indoor steel cages (either pairwise in 0.5 m^2^ cages or in family cages of 1 m^2^, 2.5 m height each, fulfilling the requirements of Directive 2021/63/EU of the European Union) either with a male or female partner animal or in a family group. Cages were equipped with one sleeping box per two animals, a hammock, branches, seat boards, wooden chips as litter on the ground, and weekly alternating enrichment such as swings and cartons. Temperature in the enclosures was kept constant at 26 ± 1 °C, with a relative humidity of 60–80% and 10 air changes per hour. Circadian rhythm was maintained at an alternating 12 h duration of light and dark. The diet was composed of a portion of porridge containing curd or milk mash and additional vitamins, minerals, and proteins in the morning. For lunch, animals were given a mixture consisting of fruit, vegetables, and pellets (ssniff Mar, ssniff Spezialdiäten GmbH, Soest, Germany). The daily diet was variated by adding, for example, mealworms, gum arabic, nuts, seeds, and eggs. Water was offered in drinking bottles ad libitum. An enrichment protocol with, e.g., varying food and environmental enrichment, was implemented as part of the animal husbandry. Seven healthy test-naive male common marmosets (*Callithrix jacchus*) were included in this study. Animals were selected by the husbandry unit of the German Primate Center, Göttingen, Germany. Before incorporating the animals into the study, a clinical and hematological assessment was carried out to ensure their health status. The mean age of 32.83 ± 1.08 months at the beginning (MRI measurement) reflects a human age of approximately 25–30 years. At this time, animals weighed, on average, 435 g ± 11.55 g (SEM). For characterization of the animal cohort at the time of PV loop measurement, see Table 2.

### 2.3. Study Design

All animals underwent MRI to assess cardiac function and heart morphology. After a period of at least four weeks, PV loop measurements were performed. One of the seven animals had to be excluded due to arrhythmia from the PV loop analyses (see Appendix A). Additionally, five animals were selected for echocardiography measurements directly prior to the PV loop measurements. These circumstances resulted in a total of seven animals for MRI analysis, six animals for PV loop analysis, and five animals for echocardiography. For direct comparison between the three methods, only data from animals for which all three datasets could be acquired were analyzed.

Subsequently, after PV loop measurements, animals were euthanized under deep anesthesia with an overdose of potassium chloride (>75 mg/kg BW, Kaliumchlorid 7.45%, B.Braun, Melsungen, Germany), instead of barbiturate, to prevent crystalline deposits in the heart tissue. Major organs including the heart were excised during necropsy and used for further analyses unrelated to this study.

No blinding was performed during MRIs, echocardiography, or PV loop measurements to ensure correct identification of the respective animals. Data were analyzed in a blinded manner.

### 2.4. MRI

Induction of anesthesia was achieved by intramuscularly injecting alfaxalone (10 mg/kg, Alfaxan, Jurox, Australia) and diazepam (0.125 mg/animal, Diazepam-ratiopharm, Ratiopharm, Ulm, Germany). After placement of a 26G intravenous catheter (dual entry, Deltaven, FEP, DELTA MED, Viadana, Italy) into the vena saphena or the caudal vein and intubation of the animals (self-made endotracheal tube, inner ⌀ 1.5 mm, outer ⌀ 2.1 mm), anesthesia was maintained with 2% propofol (36–60 mg/kg/h, Narcofol, cp-pharma, Burgdorf, Germany) and remifentanil (10–25 µg/kg/h, Remifentanil Kabi, Fresenius Kabi, Bad Homburg vor der Höhe, Germany). Animals were mechanically ventilated (respiration rate 35/min, Animal Respirator Advanced 4601-2, TSE Systems, Berlin, Germany). Isoflurane (0.6–2.0%, Isofluran CP, cp-pharma, Burgdorf, Germany) in a mixture of oxygen (50%) and ambient air (50%) was only used additionally to support or increase the depth of anesthesia. The flow rate was set to 1.5–2 l/min. To protect eyes from drying out, eye ointment (Bepanthen AS, Bayer, Leverkusen, Germany) was applied to the cornea. Vital parameters such as heart rate, peripheral oxygen saturation (pulse oximetry), respiration rate, ECG, and rectal temperature were continuously monitored during the measurement (ERT Control/Gating Module 1030, SA instruments, Maharashtra, India).

Anesthetized animals were placed in a sphinx position (see Figure 2A). Ear bars, dipped with lidocain ointment (Emla 5%, AstraZeneca, Cambridge, UK), served as hearing protection. Body temperature was kept at a constant level by using warm water-filled blankets and small isolating cushions covering the animal. For antiemesis, maropitant (1 mg/kg, Cerenia, zoetis, Berlin, Germany), and, for anti-inflammatory analgesia, meloxicam (0.2 mg/kg, Metacam, Boehringer Ingelheim, Ingelheim am Rhein, Germany) were subcutaneously given during recovery period.

MRI experiments were performed on a 9.4 Tesla small animal MRI system (BioSpec 94/30, Bruker BioSpin MRI GmbH, Ettlingen, Germany) equipped with a 330 mT/m gradient system (BGA-20S, Bruker BioSpin MRI GmbH, Ettlingen, Germany). For signal reception, a single-loop receive coil with elliptical shape and diameter of 38 mm × 35 mm (O-HLE-94, Rapid Biomedical, Rimpar, Germany) was used. The ventricular function was quantified, as previously described [22], using short-axis images with whole heart coverage acquired during free breathing using a navigator-based IntraGate-FLASH sequence with following parameters: 40 navigator points, repetition time (TR) = 10.5 ms, echo time (TE) = 1.6 ms, flip angle = 35°, field of view (FOV) = 51.2 × 51.2 mm^2^, acquisition bandwidth = 200 kHz, echo position = 33.33%, spatial resolution = 0.2 × 0.2 × 1.2 mm^3^, 2 × 2 in-plane interpolation, slice gap = 0.3 mm, 12 slices, and 200 repetitions, resulting in a total acquisition time of approximately 54 min (see Figure 2B,C). After manual myocardial segmentation using Medviso Segment (Version 2.0 R6435, Medviso, Lund, Sweden) [34], the ventricular function was assessed using Simpson’s method of disks [35].

### 2.5. Echocardiography

Initial anesthesia was achieved by intramuscularly injecting alfaxalone (10 mg/kg, Alfaxan, Jurox, Australia) and benzodiazepine (either diazepam (0.125 mg/animal, Diazepam-ratiopharm, Ratiopharm, Ulm, Germany) or midazolam (0.25 mg/animal, Midazolam-ratiopharm, Ratiopharm, Ulm, Germany)) into the quadriceps muscle. A 26G intravenous catheter (dual entry, Deltaven, FEP, DELTA MED, Viadana, Italy) was inserted into the vena saphena or alternatively the tail vein. Eye ointment (Bepanthen AS, Bayer, Leverkusen, Germany) was used to prevent eyes from drying out. Abdomen, thorax, and neck were shaved with small electronic clippers up to the junction line of the mandibular branches.

For transthoracic echocardiography, a sonography system (Vevo 3100, Software Version 3.1.1.13103) and corresponding transducer (MX 250 or 400, 15–30 MHz or 20–46 MHz, respectively) with ultrasound gel were used. Animals were secured with adhesive tape in supine position on a heated plate (38–39 °C) with integrated electrocardiogram electrodes (see Figure 2D). A temperature sensor was rectally inserted for continuous monitoring of body temperature. The heart action was recorded in B-Mode in left parasternal long-axis view by averaging analyses of three consecutive heart cycles. Volumetric calculations were performed using Simpson’s monoplane method (see Figure 2E,F).

### 2.6. PV Loop Measurement

Subsequently, after conducting the ultrasound, the animals were intubated (self-made tubes, inner ⌀ 1.5 mm, outer ⌀ 2.1 mm) and repositioned on a surgery table. Electrocardiography electrodes were attached to shaved points on the outer site of both forearms and one thigh. Anesthesia was maintained with 2% propofol (36–60 mg/kg/h, Narcofol, cp-pharma, Burgdorf, Germany) and remifentanil (10–25 µg/kg/h, Remifentanil Kabi, Fresenius Kabi, Germany) via continuous infusion (Perfusor: Perfusor Space, B. Braun Deutschland GmbH & Co., KG, Melsungen, Germany) and supported by a pressure-controlled ventilation with 1.5–3% sevoflurane (SevoFlo, zoetis, Zaventem, Belgium) in a mixture of 40% oxygen and 60% ambient air. Sevofluran doses were adjusted according to the depth of anesthesia, with a constant ventilation flow of 2 L/h and post end-expiratory pressure of 1–4 mmHg (MAQUET Flow-i, Getinge Deutschland GmbH, Rastatt, Germany). Depth of anesthesia was assured by checking on reflexes such as corneal reflex and inter-toe reflex as well as continuously monitoring vital parameters such as heart rate, peripheral oxygen saturation (pulse oximetry), respiration rate, and rectal temperature (iPM12 Vet Touchscreen Surveillance Monitor, Shenzhen Mindray Bio-Medical Electronics Co., Ltd., Shenzhen, China). During the intervention, body temperature was maintained at a physiological level by using a flat infrared heating pad (38–39 °C) and an inflatable thermal blanket (temperature option 42 °C, MistralAir with Cover: Paediatric Underbody Plus (MA2475); both: The 37 °C Company, The Surgical Company International B.V., AJ Amersfoort, The Netherlands). The animals were placed in supine position on a rubber mat with the heating pad underneath and the thermal blanket above it, with arms and legs stretched outwards and fixated with a gauze bandage, if necessary.

#### 2.6.1. Surgery for Hemodynamic Measurement Intervention

Opening the thorax made it necessary to ligate the trachea to the non-cuffed tube beforehand to secure optimal ventilation. Therefore, after local anesthesia (Xylocain Gel 2%, Aspen Pharma Trading Limited, Dublin, Ireland), a median ~3 cm incision of the skin via blunt dissection was performed directly above the trachea, starting shortly below the larynx. Small elastic retractors (3311-1G, LoneStar Retractor System, CooperSurgical, Trumbull, CT, USA) were attached to create a clear view of the surgery field. After this step, local anesthesia (carbostesin, Aspen Pharma Trading Limited, Dublin, Ireland) was applied to the muscle layer. Muscle and fat around the trachea were gently prepared aside without damaging the surrounding tissue. The trachea was then ligated to the non-cuffed tubus (see Figure 2G) with non-absorbable braided silk suture, size 6/0 (Fine Science Tools, Heidelberg, Germany).

For hemodynamic measurements, an abdominal approach was selected (see Figure 2H–J): the abdominal wall was shortly lifted, caudal to the xiphoid process, and opened toward both sides following the thoracic arch with delicate sharp/blunt scissors (Cat. BC 112 R, B.Braun Melsungen AG, Melsungen, Germany), sparing the underlying organs. The diaphragm was opened following the same procedure, sparing the underlying lung. In order to perform vena cava occlusions, a loose ligation was placed around the inferior vena cava (see Figure 2I, arrow indicating vena cava inferior). Finally, the pericardium was opened at the apex of the heart and gently stripped aside with cotton tips. Access for the PV loop catheter (Scisense Catheter 1.9F, Transonic Europe B.V., Elsloo, The Netherlands) was created by puncturing the left ventricle from the apex of the heart with a 24G cannula directed to the aortic valves, but not more than 0.5 cm to ensure central positioning of the catheter in the ventricle. After removing the cannula, the catheter was inserted into the left ventricle via this tunnel (Figure 2J). After placement of the catheter, the surrounding surgery field was covered with warm and wet cotton pads to prevent unnecessary temperature drop and fluid loss via the body surfaces. Appendix B provides additional considerations on performing PV loop measurements in the common marmoset monkey.

#### 2.6.2. Hemodynamic Measurement

Before the pressure–volume measurement some preparation steps had to be performed:(a)Blood and heart muscle resistivity were determined for each animal using a calibration probe (5.0 mm, Model FM-1287-IM, Transonic Europe B.V., Elsloo, The Netherlands). For measuring blood resistivity, ~0.2 mL blood was drawn from the femoral vein just before the intervention and added to a 1.5 mL Eppendorf tube. The calibration probe was immediately introduced to the blood to acquire values for resistivity. For determination of the heart muscle resistivity, the calibration catheter was positioned directly on the heart surface avoiding coronary arteries, before inserting the cannula as described above. All readouts were directly saved into the ADV500 system.(b)The pressure–volume catheter was presoaked in body-warm saline in a 1 mL syringe on the level of the animal’s heart 20 min prior to the measurement to adjust to 37 °C fluid. After zeroing pressure offset, the catheter was inserted into the ventricle lumen.

The catheter was positioned in the center of the left ventricle by confirmation of the correct shapes of the pressure–magnitude loops (magnitude versus pressure) and the phase signal. Followed by a baseline scan to determine Gb-ED and Gb-ES, the measurement was started, and absolute blood volume and pressure loops were collected. Sampling rate was set to 400/s, and smoothing or digital filtering (30–50 Hz) was applied to filter out interference from other equipment, if necessary. To acquire a set of loops without potential breathing artifacts, ventilation was switched off for some seconds.

To measure load-independent parameters, vena cava occlusions were conducted by carefully lifting the placed thread for a few seconds, until appropriate pressure and volume reductions could be observed (see Figure 2K,L for corresponding PV loops).

#### 2.6.3. Analysis of PV Loop Data

Data collected with ADV500 Pressure-Volume Measurement System (Transonic Europe B.V., Elsloo, The Netherlands) were analyzed using LabChart 8 Pro software (Version v8.1.16 12.12.2019, ADInstruments, Sydney, Australia).

Per animal, a section of 13–35 representative loops for baseline data and 9–14 representative loops for occlusion data, depending on heart frequency as well as loop quality and steadiness, were selected and analyzed. Calculation for stroke volume (SV), ejection fraction (EF), cardiac output (CO), and arterial elastance (Ea) were performed by using the following formula (EDV = end-diastolic volume; ESV = end-systolic volume; ESP = end-systolic pressure; Ees = end-systolic elastance; EDP = end-systolic pressure):SV (µL) = EDV − ESV
EF (%) = SV/EDV
CO (µL/min) = heart rate (bpm) × SV
Ea (mmHG/µL) = ESP/SV

Tau (isovolumic relaxation constant) was calculated by LabChart program using the Weiss method.

For analysis of the compliance and contractility parameters, generated by performing vena cava occlusions, equations, as given in LabChartPro program, were used to calculate the respective parameters. For end-systolic pressure–volume relationship (ESPVR), parameters were calculated with linear equations as well as quadratic equations; for end-diastolic pressure–volume relationship (EDPVR), linear and exponential fits were applied to gain a comprehensive overview of possible collectable values. Respective coefficients of determination (r^2^) were reported, as given in LabChartPro program.

ESPVR linear fit:ESP = Ees × ESV + pressure axis intercept
V100 = calculated volume at a pressure of 100 mmHG

ESPVR quadratic fit:ESP = a × (ESV)^2^ + b × (ESV) × c
a = coefficient of curvilinearity
b = stiffness constant

EDPVR linear fit:EDP = dp/dV × EDV + pressure axis intercept
dp/dV = slope of EDPVR, index of stiffness

EDPVR exponential fit:EDP = a^(b × EDV)
a = k2-EDPVR = constant
b = k1-EDPVR = dp/dV = stiffness constant

The potential energy (PE) and pressure–volume area (PVA) were calculated by LabChartPro using linear equation models for both ESPVR and EDPVR. For preload recruitable stroke work (PRSW), the linear equation, as given in LabChartPro, was used.

### 2.7. Statistics

Statistical analysis was conducted using SigmaStat 4.0 (Systat Software Inc., Düsseldorf, Germany). Graphs were plotted using GraphPad Prism 9.3.1 (GraphPad Software, San Diego, CA, USA). Normal distribution of all parameters was confirmed using the Shapiro–Wilk and Kolmogorov–Smirnov tests. Descriptive statistics were applied. Quantitative values are stated as mean ± standard error of the mean (SEM).

For comparison of the methods, HR, EDV, ESV, EF, SV, and CO were assessed with each method. Repeated-measurements ANOVA followed by Bonferroni adjustments were used to compare the parameters.

*p*-values <0.05 and <0.01 were considered as significant and highly significant, respectively. *p* < 0.001 was defined as being highly significant.

## 3. Results

### 3.1. PV Loop Measurements

An overview of the PV loop data and the corresponding graphs is presented in Table 3 and Figure 3. The heart rate (HR) for the six selected animals was, on average, 198 ± 14 beats per minute (bpm). With a mean end-diastolic volume (EDV) of 315 ± 27 µL and a mean end-systolic volume (ESV) of 89 ± 21 µL, the stroke volume (SV) amounted to 226 ± 25 µL. This resulted in a mean cardiac output (CO) of 44 ± 6 mL/min and a left ventricular ejection fraction (EF) of 72 ± 6%. The stroke work (SW) amounted to 9191 ± 768 mmHG∗µL. The maximum rate of pressure change (dP/dt max) in the ventricle was 1429 ± 166 mmHG/s, and the minimum pressure change (dP/dt min) accounted for −1221 ± 142 mmHG/s. The arterial elastance (Ea) was 0.23 ± 0.03 mmHG/µL. For the isovolumic relaxation constant Tau, a mean value of 19 ± 1 ms was given.

For the assessment of contractility and compliance parameters, vena cava occlusions were performed. As a first-time description of PV loop measurements, parameters derived from both linear and quadratic equation models for the end-systolic pressure–volume relationship (ESPVR) as well as from both linear and exponential equation models for the end-diastolic pressure–volume relationship (EDPVR) are presented (also see Table 3). The mean coefficients of determination (r^2^) for ESPVR were, on average, 0.816 ± 0.069 for the linear fit and 0.857 ± 0.048 for the quadratic fit. For EDPVR, an r^2^ value of 0.867 ± 0.035 was calculated for the linear fit and 0.843 ± 0.039 for the exponential fit. The end-systolic elastance (Ees), derived from the linear equation for ESPVR, was 0.467 ± 0.095 mmHG/µL. The volume at a pressure of 100 mmHg (V100) was calculated to be 214 ± 55 µL, using the linear equation model. The constant of curvilinearity (a) amounted to −0.012 ± 0.004 for the quadratic fit of ESPVR. The constant determined by nonlinear least squares regression (b) was 1.634 ± 0.494. For EDPVR, the index of stiffness (dp/dV)—the slope of the EDPVR—was 0.027 ± 0.009 mmHG/µL for the linear fit and 0.013 ± 0.004 mmHG/µL for the exponential fit. The axis intercept of the linear equation for EDPVR amounted to −1.132 ± 1.189. The constant k2 was 0.793 ± 0.292 in the exponential curve. Preload recruitable stroke work (PRSW) was described by a linear fit model with a mean r^2^ of 0.945 ± 0.017. The slope of the PRSW, representing the relation of SW and ESV and serving as an index of contractile function, was, on average, 40 ± 5 mmHG. The axis intercept for PRSW was, on average, −1949 ± 506 mmHG. For myocardial energetics, the elastic potential energy (PE) was calculated with linear models for both ESPVR and EDPVR and amounted to 3112 ± 1181 mmHG∗µL. This resulted in a mean pressure–volume area (PVA) of 6217 ± 1689 mmHG∗µL (also see Table 3 and Figure 3).

### 3.2. MRI Measurements

Volumetric and functional parameters of the left ventricle were obtained by MRI (free breathing IntraGate Flash in the short-axis orientation with whole heart coverage). The HR (average: 225 ± 17, range: 139–272 bpm) during the MRI exam widely varied among the animals. However, the assessed volumes only showed mild variations within the expected heart-size-dependent differences. In detail, the average values of the left ventricular mass (LVM), EDV, ESV, and SV were 1.21 ± 0.04 g, 956 ± 53 µL, 365 ± 15 µL, and 591± 46 µL, respectively. The resulting EF was, on average, 61 ± 2%. In addition, investigations of the right ventricular function were performed: right ventricular mass (RVM), right ventricular end-systolic volume (RV-ESV), right ventricular end-diastolic volume (RV-EDV), right ventricular stroke volume (RV-SV), and right ventricular ejection fraction (RV-EF) were measured and accounted for 0.57 ± 0.02 g, 356 ± 32 µL, 875 ± 45 µL, 518 ± 28 µL, and 59 ± 2%, respectively (Figure 4).

### 3.3. Echocardiographic Measurements

The echocardiographic investigations of the left ventricle in long-axis view also allowed for measuring ESV, EDV, SV, HR, CO, and EF, as with MRI and PV loops. In addition, left ventricular internal diameter in systole (LV IDs) and in diastole (LV IDd) as well as fractional shortening (FS) could be measured. The speckle tracking method of the echocardiographic system also allowed for strain analyses of the left ventricle. Accordingly, time to peak and peak could be acquired for the longitudinal and radial velocity, longitudinal and radial displacement, longitudinal and radial strain, and longitudinal and radial strain rates. The values for these parameters are presented, from the midventricular anterior wall segment, in Figure 5.

### 3.4. Comparison of PV Loops with MRI and Echocardiography

Prior to PV loops, MRI measurements were conducted in all animals and, additionally, echocardiographic analyses were performed in five of the animals. Comparisons of the volumetric and functional parameters of the left ventricle obtained by MRI (free-breathing IntraGate Flash in the short-axis orientation), PV loops, and ultrasound for the same five animals are summarized in Figure 6 and Appendix A. The PV loop and ultrasound resulted in comparable values for EDV, ESV, SV, and, consequently, CO (*p* > 0.05, Figure 6). In comparison, MRI analyses showed significantly different values for these parameters: EDV, ESV, and SV were each approximately three- to four-fold higher than when measured with the other two methods, whereas HR and EF were similar. This also resulted in a three- to four-fold higher CO calculated from MRI. Interestingly, although showing the highest values for the absolute volumes, the variability of the data (as assessed via coefficients of variations) was usually lowest for MRI (data not shown).

## 4. Discussion

Given the increasing interest in the common marmoset as an in vivo model in various research areas [7,8], a comprehensive characterization of this species’ cardiovascular system is urgently needed.

The use of NHPs like common marmosets raises some ethical concerns [36] and they have, in general, a lower availability than small rodents for research. Additionally, the common marmoset differs from the human situation with regard to its higher heart rate and slight differences in cardiac anatomy [37]. The facts that the possibilities for genetic modifications are currently not yet as advanced as for rodent species and that there are fewer amounts of available cross-reacting antibodies did limit their broader use in comparison to other animal species in the past [38,39]. However, common marmosets exhibit a relative heart weight, general cardiac anatomy, and mean myocyte volume similar to humans; they show spontaneous development of heart fibrosis, inflammatory cell infiltration, and myocardial degeneration as well as similar age-related changes in microvasculature and atherosclerosis [19,21,40,41,42,43]. Additional similarities to humans in the field of metabolism (e.g., the induction of obesity and diabetes is spontaneously or experimental possible and similar lipoprotein profiles) [14], immunology, and aging (e.g., neurological and cognitive decline and cartilage changes) [15,17] predispose this species as a promising animal model for biomedical research, inter alia investigations of heart failure, and comorbidities of cardiac diseases. Appendix A shows a broad overview over commonly used animal species and their respective advantages and disadvantages for cardiovascular research.

The present study not only provides an anesthesia and surgery protocol for the acquisition of pressure–volume hemodynamic data but also reveals, for the first time, basic values from young male marmosets as references for future studies. The comparison to the established cardiovascular assessment methods, echocardiography and MRI, as well as the discussion of the respective advantages and disadvantages supports researchers in deciding which method to use for their specific scientific question.

### 4.1. Study Design

PV loop measurements are considered as the standard to comprehensively assess cardiac function in animal models due to their ability to measure highly sensitive alterations in left ventricular function [27]. In general, two approaches exist for PV loop measurements: using the closed chest approach, a PV catheter is inserted via the carotid or femoral artery and retrogradely advanced over the aortic valve into the left ventricle. In contrast, for the open-chest approach, which was used in the present study, the abdominal cavity has to be opened, followed by an incision into the diaphragm. Subsequently, the catheter is inserted into the heart after puncturing the left ventricle apex. Both techniques have advantages and disadvantages: with the closed chest approach, the lung ventilation, intra-cavity pressure values, and heart position are more in accordance with the physiological situation. In direct comparison to both approaches in mice, EF and SV are higher with the closed chest approach (79% and 19.6 µL versus 66% and 15.6 µL, respectively, in the closed chest approach). dP/dt min and Tau are reduced, whereas ESPVR, EDPVR, and PRSW usually remain unchanged [27]. However, the assessment of load-independent parameters like ESPVR or EDPVR is more reliable by performing vena cava occlusions in the more invasive setting of abdominal approaches. Usually, the open-chest approach for PV loop measurements is well-accepted for these reasons and is, therefore, often considered the gold standard to comprehensively acquire heart functions in rodents. In the beginning of this study, the carotid approach was attempted but, due to anatomical specialties in the common marmoset’s aortic arch, the catheter could not be advanced into the left ventricle and measurements could not be performed (data not shown [19,44]). Therefore, the abdominal approach was selected.

### 4.2. PV Loop Data in the Context of Reported Cardiovascular Data for the Common Marmoset

As a first-time description, the basic data of PV loop measurements for young, healthy male marmosets were provided. Since admittance-based PV loop studies had not been performed in this animal species before, any comparison to previous studies is complicated: the group of McLennon et al. calculated the cardiac parameters for common marmosets based on radionuclide angiography data during a feeding experiment 30 years ago [45]. Their investigations showed roughly similar volume data for EDV (~353 µL) and SV (~170 µL), as reported in the present study. However, EF (~48.5–59.8%) was lower, while ESV (~183 µL) was higher, than in this study. Differences in animal selection (they used younger and lighter individuals of 12–15 months and, on average, 323 g), study design (anesthesia with pentobarbitone sodium in their study; HR with 280 ± 25 bpm was considerably higher), and technical differences (the calculation of cardiac parameters from radionuclide angiography) might explain the contrasting results. Another study previously reported on echocardiographic data for common marmosets in which a combination of ketamine/medetomidine was used for sedation prior to assessing heart function [46]. Their values were 1.57 ± 0.09 mL for EDV and 0.5 ± 0.05 mL for ESV, respectively, thereby even exceeding the presented MRI data. However, the EF of 67% was similar to the values reported here. Unfortunately, the authors did not report on HR, and the differences in measurement (left apical axis versus left parasternal axis) and anesthesia protocol (ketamine and medetomidine are known to significantly affect cardiac functions) complicate comparisons between the methods.

To the best of the authors’ knowledge, only one other study (performed by coauthors of the current manuscript) reports on the cardiac MRI values of common marmosets. Moussavi et al. investigated the cardiovascular aging process of this animal species via MRI and observed values similar to those presented here [22]. The reported data are in line with the anatomical descriptions of and volume estimations for this species [21]. Furthermore, when compared to other animal species of similar sizes like rodents or squirrel monkeys, the acquired data were also in line with previous reports [47,48,49,50].

### 4.3. Comparison of PV Loop Measurements with Echocardiography and MRI

For the cardiovascular parameters, which could also be collected via echocardiography and MRI, similar values of HR, EDV, ESV, SV, and CO for PV loops and echocardiography were assessed, whereas the volume MRI data were significantly higher. However, although differences in absolute values could be observed, EF remained rather similar between the methods. Interestingly, although showing the biggest variations in absolute values, the coefficients of variations for MRI were often lower compared to those for echocardiography and PV loops (data not shown).

The literature research on comparisons of different imaging methods resulted in a mixed picture: studies in larger animal species or humans report on higher volumes in MRI and either slightly lower or higher EFs compared to echocardiography [51]. In contrast, PV loops previously resulted in similar to slightly higher volume values compared to MRI, whereas EF was stable [28,52]. Apart from these differences, the methods usually show good correlations to each other in bigger species [53,54]. Similarly, the volumetric assessments of cardiac functions in smaller animals like mice previously showed slightly higher, similar, or lower values between echocardiography and MRI, whereas EF was either slightly higher or again similar between methods [55,56,57]. Of particular interest is one study by Jacoby et al. in 2014, in which the group conducted cardiac MRIs directly following PV loop measurements via the open-chest approach in mice. Similar to the present study, they also gathered three- to four-fold higher volumetric values for MRI, whereas EF was similar in direct comparison [58]. Not only the technical differences between both methods (e.g., the value calculation differences between methods and catheter insertion in the ventricle) but also the invasiveness of open chest PV loops, accompanied by pressure changes in the thorax with resulting physiological consequences for the pressures and volumes, might be important reasons to be considered when deciding on the method of choice. On the other hand, whereas agreement between methods should be better in bigger animal species and humans, the small sizes of the cardiac chambers in combination with the higher heart rates in smaller animals may lead to a challenging spatial and temporal resolution and differences in the total volumes acquired via imaging techniques like MRI compared to other methods [59]. Combined with differences in the position of the animals (the sphinx position in MRI and the supine position in echocardiography and PV loops), the calculation methods for each parameter (Simpson’s method of disks for MRI, Simpson’s monoplane for echocardiography, and admittance-based calculations for PV loops), and technical prerequisites, the contrasting results between methods could be explained. Although the anesthesia regimens were comparable between methods (induction with benzodiazepines and alfaxalone, maintenance with propofol and remifentanil, and supplementation with either sevoflurane for PV loops or isoflurane in low concentrations for MRI), these differences could potentially have contributed a small amount to the observed findings.

The inconsistencies in the results could also be partially explained by the different time points for the respective measurement methods. Through the use of machine learning, artificial intelligence (AI) could help identify the sources of error that arise from the temporal differences in measurements. By analyzing the available data and accounting for temporal variations, AI could identify patterns that are consistent over time. In this way, AI possibly aids in the evaluation of the cardiac function assessments described above and, thus, may complement previously described applications in the prevention and diagnosis of heart failure [60].

### 4.4. Considerations for PV Loop Measurements in Common Marmosets

Not only the surgical approach can significantly influence the outcome of hemodynamic measurements. One important factor is the anesthetic regimen: for the least cardio-depressive effects, the induction of anesthesia was performed with alfaxalone and benzodiazepines, followed by a combination of propofol and remifentanil supplemented with sevoflurane for maintenance, to provide balanced anesthesia with simultaneous adequate analgesia. The comparable low pressures during PV loop measurements indicate good anesthetic depth, which was confirmed by the maintained good peripheral blood perfusion. However, studies for the optimization of the anesthesia protocol could benefit the outcomes of future PV loop measurements. In addition to the anesthetic regimen, factors like body temperature and hydration status are known to influence cardiac performance. Similar to other small animal species, the common marmoset is also susceptible to temperature loss and dehydration under anesthesia, especially when invasive procedures are performed [61]. Considerable care was, therefore, given to ensure the adequate temperature and hydration management of the animals.

Of the seven initially planned animals for this study, one animal had to be excluded from the hemodynamic measurements due to severe fibrillations as soon as the PV loop catheter was inserted into the heart. This unusual observation might be due to an underlying susceptibility of individual animals to the electric field emitted by the catheter. However, this phenomenon should be confirmed in future investigations.

In conclusion, it was shown that PV loop measurements can be reliably performed in the common marmoset, and they nicely complement standard routine cardiological measurements like ultrasound and MRI. PV loops offer the additional benefit of providing pressure–volume relationships and give indications about ventricular chamber mechanical properties like contractility and chamber compliance, ventricular arterial coupling, and myocardial energetics. In contrast, ultrasound and MRI enable the repeated assessment of basic cardiovascular parameters and, in the case of MRI, provide the possibility of also measuring perfusion kinetics. Whereas the different methods might result in differences for absolute values, all of them reliably provide functional parameters. In addition to already established cardiovascular diagnostic methods, the measurement of hemodynamic functional parameters via PV loops, therefore, represents a valuable tool for the comprehensive assessment of cardiovascular function and allows for the assessment of important information about the common marmoset’s cardiovascular system in health as well as under pathological conditions.

## 5. Conclusions

In summary, it was shown that all three commonly used cardiovascular assessment techniques can be performed in the common marmoset, each with its own advantages and disadvantages: echocardiography and MRI offer, as non-invasive imaging techniques, the possibility to repeatedly assess the heart function over the course of a study. However, especially MRI measurements are time-consuming and require considerable technical efforts; although important parameters can be assessed, information about intra-cardiac pressures cannot be obtained. Especially in small animals, it can be challenging to precisely determine volumetric data due to small chamber sizes, turbulent blood flows, and short T2 relaxation times and valve movements and, therefore, requires a specialized MRI team and technical equipment. However, up to date, MRI measurements have been considered the gold standard to precisely and non-invasively detect cardiac volumes. Contrast-enhanced protocols allow for the assessment of myocardial infarction in the MRI setting, and MR spectroscopy enables the study of the cardiac metabolism.

Echocardiography is commonly available and can be routinely used to assess parameters like the wall and chamber dimensions. It requires appropriate expertise with imaging modalities; otherwise, the acquired values may be variable. The possibility to perform deformation imaging in the form of myocardial shortening provides information on the contractility and regional wall motion. Echocardiographic and MRI volumetric measurements usually rely on volumetric assumptions, and volume calculations require post-processing.

In contrast to MRIs and echocardiography, PV loop measurements allow for comprehensive measuring, mostly of load-independent parameters, in real time, including the unique possibility to assess the LV performance contractility, elastance, energetics, and efficiency—all important to adequately assess heart function, especially under pathological conditions. Importantly, one has to be careful that the catheters are properly calibrated to accurately assess the volumetric values.

Overall, echocardiography and MRIs are useful for non-invasive serial measurements, with the appropriate expertise and technical equipment. Invasive PV loop-measurements, in contrast, allow for a more comprehensive estimation of cardiac function, with the advantage of assessing the parameters on a beat-to-beat basis.

## Figures and Tables

**Figure 1 biology-12-01123-f001:**
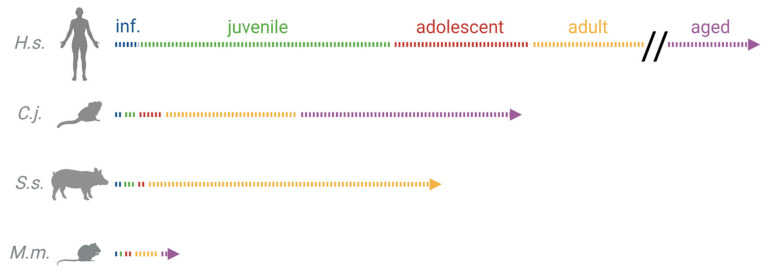
Life span of humans and different animal models in comparison. inf. = infant; *H.s. = Homo sapiens*; *C.j. = Callithrix jacchus*; *S.s. = Sus scrofa domesticus*; *M.m. = Mus musculus*. Created with Biorender.com.

**Figure 2 biology-12-01123-f002:**
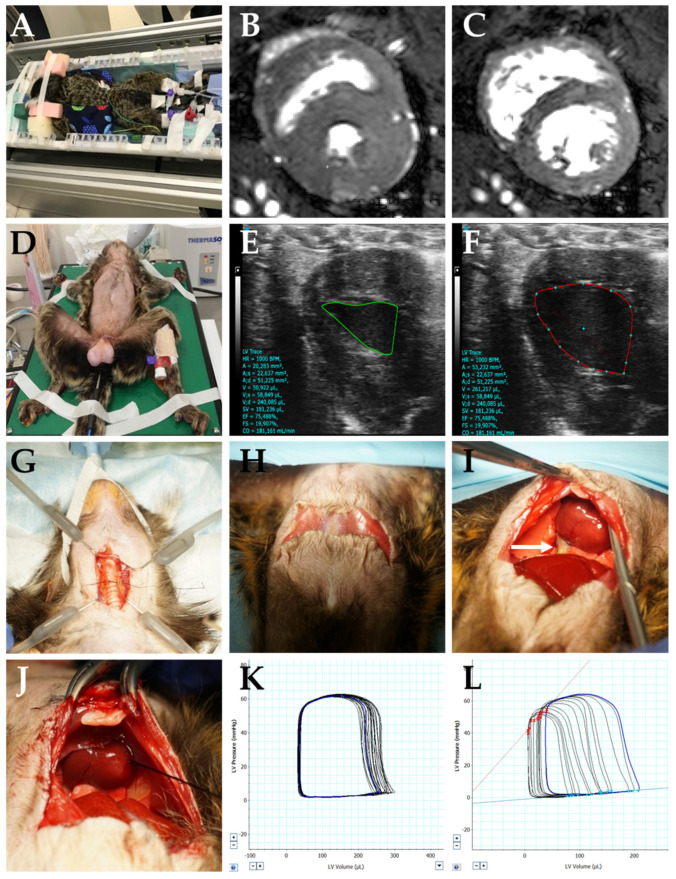
Steps of the MRI measurement, echocardiographic examinations, and surgery for hemodynamic pressure–volume measurements. (**A**–**C**) MRI measurements: setup for MRI measurements including intubation, vital parameter monitoring, venous access, and warming pads for temperature management (**A**) and exemplary MRI short-axis view in systole (**B**) and diastole (**C**). (**D**–**F**) Setups for echocardiographic investigations: the animal was placed on a heating mat capable of monitoring heart rate and respiration rate, an additional temperature probe was inserted into the rectum (**D**), and echocardiographic image left parasternal long-axis view in systole (**E**) and diastole (**F**) was acquired. (**G**–**L**) Surgery setup for PV loop measurements: preparations of the trachea (**G**) and opening of the abdominal cavity (**H**); opening of the diaphragm, fixation of the vena cava inferior (**I**, arrow indicating vein), and insertion of the catheter into the heart (**J**); measurements of baseline PV loops (**K**); occlusion PV loops including ESPVR and EDPVR (**L**).

**Figure 3 biology-12-01123-f003:**
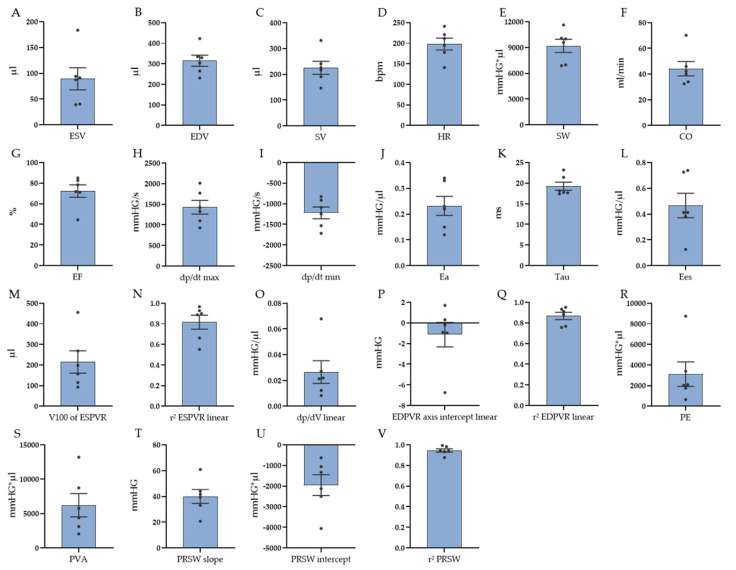
Values and variability of PV loop basic data, contractility, and compliance parameters as well as myocardial energetics. (**A**) ESV = end-systolic volume, (**B**) EDV = end-diastolic volume, (**C**) SV = stroke volume, (**D**) HR = heart rate, (**E**) SW = stroke work, (**F**) CO = cardiac output, (**G**) EF = ejection fraction, (**H**) dP/dt max = maximum derivative of pressure, (**I**) dP/dt min = minimum derivative of pressure, (**J**) Ea = arterial elastance, (**K**) Tau = isovolumic relaxation constant, (**L**) Ees = end-systolic elastance (slope of linear ESPVR), (**M**) V100 = calculated volume at a pressure of 100 mmHG end-systolic pressure–volume relationship (ESPVR), (**N**,**Q**,**V**) r^2^ = coefficient of determination, (**O**) dp/dV = index of stiffness (slope of linear EDPVR), (**P**) pressure axis intercept of linear end-diastolic pressure–volume relationship (EDPVR), (**R**) PE = elastic potential energy, (**S**) PVA = pressure–volume area, (**T**) slope of PRSW = preload recruitable stroke work, (**U**) pressure axis intercept of PRSW. Data shown as mean ± SEM (n = 6).

**Figure 4 biology-12-01123-f004:**
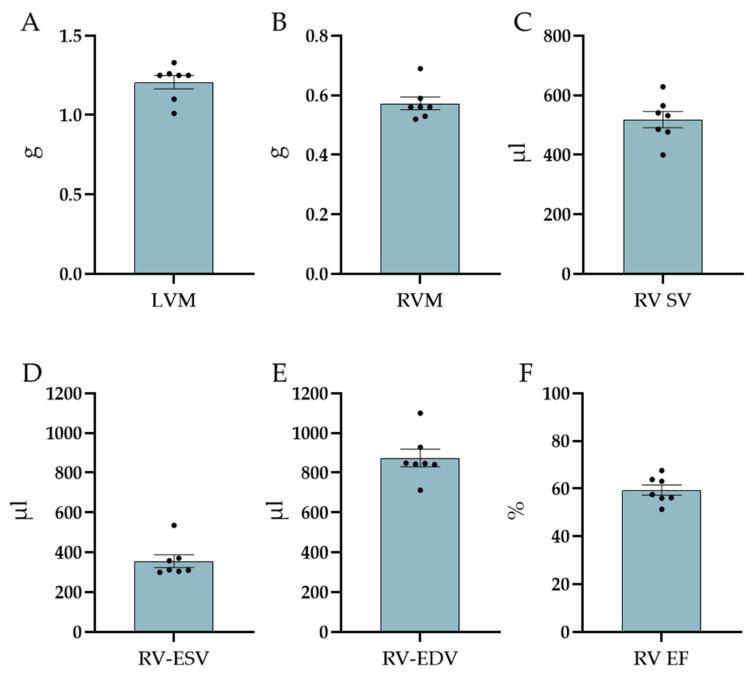
Cardiac MRI specific values—shown are: (**A**) LVM = left ventricular mass, (**B**) RVM = right ventricular mass, (**C**) RV SV = right ventricular stroke volume, (**D**) RV-ESV = right ventricular end-systolic volume, (**E**) RV-EDV = right ventricular end-diastolic volume, and (**F**) RV EF = right ventricular ejection fraction. Data shown as mean ± SEM (n = 7).

**Figure 5 biology-12-01123-f005:**
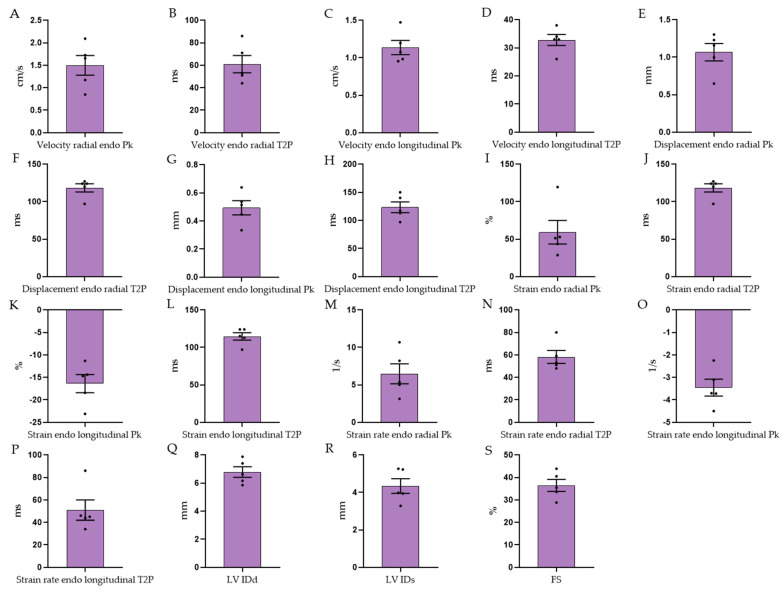
Cardiac echocardiographic specific values—shown are endocardial values for the anterior midventricular heart wall (**A**–**P**): (**A**) velocity radial peak, (**B**) velocity radial T2P, (**C**) velocity longitudinal peak, (**D**) velocity longitudinal T2P, (**E**) displacement radial peak, (**F**) displacement radial T2P, (**G**) displacement longitudinal peak, (**H**) displacement longitudinal T2P, (**I**) strain radial peak, (**J**) strain radial T2P, (**K**) strain longitudinal peak, (**L**) strain longitudinal T2P, (**M**) strain rate radial peak, (**N**) strain rate radial T2P, (**O**) strain rate longitudinal peak, (**P**) strain rate longitudinal T2P, (**Q**) representative picture of an echocardiographic strain analysis, (**R**) left ventricular internal diameter in diastole, (**S**) left ventricular internal diameter in systole, and (**T**) left ventricular fractional shortening; T2P = time to peak. Data are shown as mean ± SEM (n = 5).

**Figure 6 biology-12-01123-f006:**
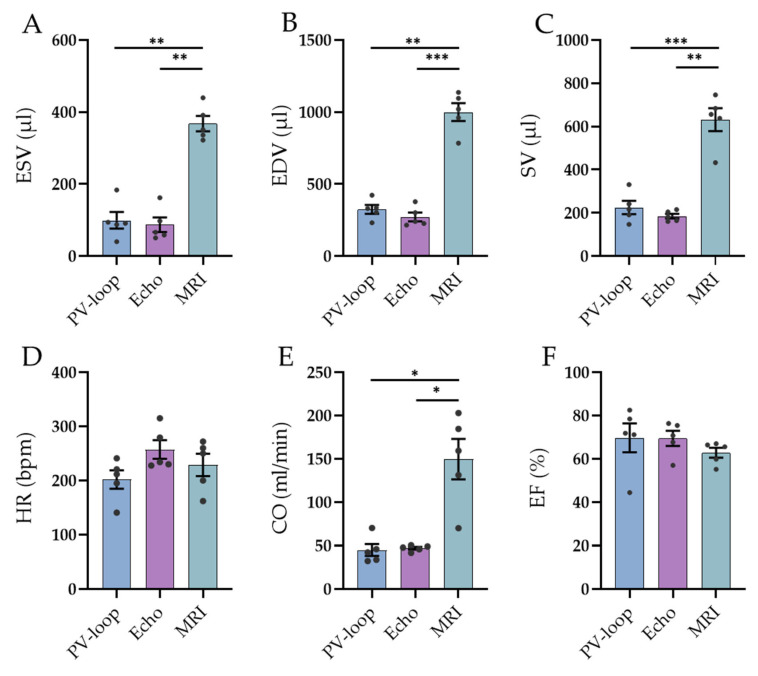
Comparison of PV loop measurements (blue), echocardiographic measurements (lilac), and MRI investigations (green). Comparison of (**A**) end-systolic volume, (**B**) end-diastolic volume, (**C**) stroke volume, (**D**) heart rate, (**E**) cardiac output, and (**F**) ejection fraction; * *p* < 0.05, ** *p* < 0.01, *** *p* < 0.001. Data shown as mean ± SEM (n = 5).

**Table 2 biology-12-01123-t002:** Basic characteristics and mean vital parameters of animals during PV loop measurements. HR = heart rate, O_2_ = oxygen saturation, RR = respiration rate, rT = rectal temperature.

Animal	Age (Months)	Body Weight (g)	Heart Weight (g)	Blood Resistivity (ρ)	Muscle Resistivity (ρ)	HR (Mean bpm)	O_2_ (Mean %)	RR (Mean/Min)	rT (Mean °C)
Animal 1	43	456	2.84	2.4	4.6	229	99	16	37.0
Animal 2	43	449	3.13	2.7	4.6	142	99	16	35.0
Animal 3	41	419	3.70	2.5	4.6	174	100	16	37.0
Animal 4	34	473	3.46	2.5	4.6	200	98	16	36.0
Animal 5	34	388	2.78	3.1	4.6	189	99	16	36.0
Animal 6	33	423	-	2.7	4.7	174	96	16	35.0
Mean ± SEM	38 ± 2	435 ± 12.5	3.18 ± 0.2	2.65 ± 0.1	4.62 ± 0.0	184 ± 11.9	98 ± 0.5	16 ± 0.1	36.0 ± 0.4

**Table 3 biology-12-01123-t003:** Mean values from selected pressure–volume measurements for baseline data as well as contractility, compliance, and myocardial energetics parameters. HR = heart rate, ESV = end-systolic volume, EDV = end-diastolic volume, SV = stroke volume, EF = ejection fraction, CO = cardiac output, SW = stroke work, dP/dt max = maximum derivative of pressure, dP/dt min = minimum derivative of pressure, Ea = arterial elastance, Tau = isovolumic relaxation constant, ESPVR = end-systolic pressure–volume relationship, PRSW = preload recruitable stroke work, PE = elastic potential energy, PVA = pressure–volume area, EDPVR = end-diastolic pressure–volume relationship, a = k2-EDPVR = constant, b = k1-EDPVR = dp/dV = stiffness constant. Data shown as mean ± SEM (n = 6).

	Parameter	Mean ± SEM (n = 6)
Baseline data		
	HR (bpm)	198 ± 14
	ESV (µL)	89 ± 21
	EDV (µL)	315 ± 27
	SV (µL)	226 ± 25
	EF (%)	72 ± 6
	CO (ml/min)	44 ± 6
	SW (mmHG∗µL)	9191 ± 768
	dp/dt max (mmHG/s)	1429 ± 166
	dp/dt min (mmHG/s)	−1221 ± 142
	Ea (mmHG/µL)	0.23 ± 0.03
	Tau (ms)	19 ± 1.0
Contractility		
	ESPVR (linear) Ees (mmHG/µL)	0.467 ± 0.095
	ESPVR (linear) V100 (µL)	214 ± 55
	ESPVR (linear) r^2^	0.816 ± 0.069
	ESPVR (quadratic) a	−0.012 ± 0.004
	ESPVR (quadratic) b	1.634 ± 0.494
	ESPVR (quadratic) r^2^	0.857 ± 0.048
	PRSW r^2^	0.945 ± 0.017
	PRSW slope	40 ± 5
	PRSW axis intercept	−1949 ± 506
	PE mmHG∗µL	3112 ± 1181
	PVA mmHG∗µL	6217 ± 1689
Compliance		
	EDPVR (linear) dp/dV (mmHG/µL)	0.027 ± 0.009
	EDPVR (linear) axis intercept	−1.132 ± 1.189
	EDPVR (linear) r^2^	0.867 ± 0.035
	EDPVR (exponential) dp/dV (k1) (mmHG/µL)	0.013 ± 0.004
	EDPVR (exponential) k2	0.793 ± 0.292
	EDPVR (exponential) r^2^	0.843 ± 0.039

## Data Availability

The data supporting the reported results of this study can be acquired upon reasonable request from the corresponding author.

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
