# Peer review of "Functional Cardiovascular Characterization of the Common Marmoset (Callithrix jacchus)"

_biology, 2023, doi:10.3390/biology12081123_

Round 1
Reviewer 1 Report
Dear authors,
you describe a well performed study to establish PV loop measurements in marmosets and subsequently compare three imaging techniques to assess cardiovascular characterization of the marmoset. In the wordfile you will find my suggestions for improvement of the manuscript. When I select minor revisions I am not able to review the manuscript again so therefore I have selected major revisions although my remarks are relatively easy to 'solve'.
Review Report:
The authors describe a well performed study to establish PV loop measurements in marmosets and subsequently compare three imaging techniques to assess cardiovascular characterization of the marmoset.
In the whole manuscript, more scientific words should be used like ‘our findings’ in line 29 and 34 could be ‘our results’. Same for line 568, 575, 592 and 754 where ‘found’ should be replaced by e.g., assessed, determined, observed, measured. ‘Chosen’ should be ‘selected’ in several lines like line 454. Line 247 ‘directly after=subsequently etc etc
Line 34: one should keep it more general like ‘primate model for cardiovascular studies’ or ‘model for cardiovascular disease’
In line 44 ‘n=5 young adult males’ but the species is missing so maybe just write, without brackets ‘…and echocardiography in five young-adult male common marmosets.’ But, however, in line 164- 184 the reader reads different numbers. Maybe write in the abstract the same numbers too so ‘seven animals for the MRI, six for the loop and five for the echocardiography. For direct comparison between the three methods, only data from animals of which all three datasets could be acquired were selected.’
Line 42: ‘aimed to’ can be deleted and change to ‘performed’.
Line 44: I would suggest to remove the whole sentence ‘after establishing……healthy animals’.
Line 47: latin names should be written in italic.
Line 75-77: delete ‘ however……turnover’ as the introduction is quite long while this sentence has no additional value as you otherwise also have to explain why not a baboon, green vervet, aotus, tamarin,….just explain why you select the marmoset for cardiac research. Subsequently, in line 78 the ‘however’ can be deleted too.
Line 81/82: delete ‘and, they can be housed …..conditions’.
Line 85-96: delete from ‘in contrast to……’ as in paragraph 1 of the introduction, the paper focus on cardiovascular diseases already. So no need to discuss all research areas of the marmoset here in detail.
Line 124: delete ‘we hope our findings’ to ‘our results will’ as you don’t hope, you are sure of your results! Delete line 126 ‘and raise…..species’.
Delete line 145/146 ‘all animals in….humanely’ as you already mention this in line 139.
Line 147: subheader should be called ‘animals, husbandry and housing’ and should include animal information like line 164-170 data. In line 165, you write ‘randomly’. I would delete ‘randomly’.
Line 148: ‘kept’ should be replaced by ‘housed’. Instead of 0.5m2 and 1m2, provide the width and depth of the cages.
Line 158: ‘food regimen was slight changed’, I assume you meant that daily diet was variated by adding….right?
Line 165, 180 and 184: ‘chosen’ should be replaced to ‘selected’.
Line 179: ad ‘due arythmia’ already here.
Line 181: ‘result‘ should be ‘resulted’
Line 185: why did you use potassium chloride and not a barbiturate for euthanasia?
Line 196: tail vein is named ‘caudal vein’. Latin names like vena saphena should be in latin (in the whole manuscript)
Line 198-203: it is was it is, but why did propofol and remifentanil need additional isoflurane as anesthetic? The flow rate is missing. Same for line 253-256: why providing double anesthesia? If you need to ventilate them to control breathing movements, you can only do isoflurane/sevoflurane so no propofol needed, right?
208: ‘figure 6 j’ should be ‘figure 2 j’. Normally, figure names should occur in line of the text so J can not be named until A-I is mentioned.
Figure 2: all images are so small that they are hard to interpret. As this is an online journal, present bigger images.
Line 230: before hemodynamic measurements was called PV loop. Keep one line otherwise it is getting confusing.
It is a suggestion: to keep clearance for the readers, one could use subheaders like
2 Materials and methods
2.1 Animals, husbandry and housing
2.2 Study design
2.3 MRI
2.4 Echocardiography
2.5 PV loop
2.6 statistics
This because the results are also divided in subheaders 3.1 PV loop 3.2 MRI 3,.3 echocardiography and in addition, it reads easier when all information about a procedure is written in one subheader.
Line 243: ‘Figure 6 G’ should be ‘Figure 2 G’ and again, mention 2G before 2A-2F is not common.
Line 270: are this the same tubes in 247-248? If yes, then one can delete it in 270.
Line 280: again, Figure 6. Please change that in the whole manuscript
Line 340: SV, EF, Co, EDV…etc are first mentioned here. It would help to have them full written instead of only abbreviated and ended in the end in a table, as a reader will think he/she missed the first time the abbreviation was used so he/she will check all pages before this one to find the abbreviation instead looking in the end of the paper. In line 381 they are full written but that is after line 340. Please adapt.
Line 381: instead of shown one could use a more scientific word like presented.
Line 394: you write ‘see also Table 2’ but I suppose this should be Table 3. In Table ‘3’ the title is also mentioned above the caption, delete. Idem line 412. Table 3 or 2?
In Line 412 I think Figure 2E and 2F are mentioned instead of Figure 2.
Figure 4 and 5 are presented before they are mentioned in the manuscript. Confusing!
Line 541: the method of choice could be ‘the gold standard’.
Line 614: “One important factor is also the choice of the anesthetic regimen” could be “One important factor is the anesthetic regimen: for……”
References: there are double references in the list e.g.
Number 10 = 49
NUMBER 13 =47
Number 20=42
Number 23=54
Number 76=50
Number 12=82
And I didn’t check them all…
Line 501 : reference 37 : this publication doesn’t discuss the availability of marmosets
Idem reference 38
In the whole manuscript, more scientific words should be used like ‘our findings’ in line 29 and 34 could be ‘our results’. Same for line 568, 575, 592 and 754 where ‘found’ should be replaced by e.g., assessed, determined, observed, measured. ‘Chosen’ should be ‘selected’ in several lines e.g., line 454. Line 247 ‘directly after'=subsequently etc etc
Author Response
Dear Reviewers,
Thank you very much for taking the time to critically review our manuscript. Your remarks and questions are highly appreciated; we did our best to address them in our revision. In the following lines, we gladly respond to your comments and elaborate on every point you kindly brought to our attention in our work:
- Reviewer 1:
In the whole manuscript, more scientific words should be used like ‘our findings’ in line 29 and 34 could be ‘our results’. Same for line 568, 575, 592 and 754 where ‘found’ should be replaced by e.g., assessed, determined, observed, measured. ‘Chosen’ should be ‘selected’ in several lines like line 454. Line 247 ‘directly after=subsequently etc etc.
We revised this text passage and the whole manuscript accordingly.
Line 34: one should keep it more general like ‘primate model for cardiovascular studies’ or ‘model for cardiovascular disease’
We revised this text passage accordingly.
In line 44 ‘n=5 young adult males’ but the species is missing so maybe just write, without brackets ‘…and echocardiography in five young-adult male common marmosets.’ But, however, in line 164- 184 the reader reads different numbers. Maybe write in the abstract the same numbers too so ‘seven animals for the MRI, six for the loop and five for the echocardiography. For direct comparison between the three methods, only data from animals of which all three datasets could be acquired were selected.’
We revised this text passage accordingly.
Line 42: ‘aimed to’ can be deleted and change to ‘performed’.
We revised this text passage accordingly.
Line 44: I would suggest to remove the whole sentence ‘after establishing……healthy animals’.
We revised this text passage accordingly.
Line 47: latin names should be written in italic.
We revised this text passage and the whole manuscript accordingly
Line 75-77: delete ‘ however……turnover’ as the introduction is quite long while this sentence has no additional value as you otherwise also have to explain why not a baboon, green vervet, aotus, tamarin,….just explain why you select the marmoset for cardiac research. Subsequently, in line 78 the ‘however’ can be deleted too.
We revised this text passage accordingly.
Line 81/82: delete ‘and, they can be housed …..conditions’.
We revised this text passage accordingly.
Line 85-96: delete from ‘in contrast to……’ as in paragraph 1 of the introduction, the paper focus on cardiovascular diseases already. So no need to discuss all research areas of the marmoset here in detail.
We thank the Reviewer for the valuable input. However, since the manuscript focuses on this new animal model in cardiovascular research, we believe this paragraph to be essential for the reader to get an overview of the common marmoset in research as well as its advantages/characteristics relevant for other areas. To follow the Reviewer’s suggestion, we shortened the whole paragraph to the most essential information and hope this finds the approval of the Reviewer.
Line 124: delete ‘we hope our findings’ to ‘our results will’ as you don’t hope, you are sure of your results! Delete line 126 ‘and raise…..species’.
We revised this text passage accordingly.
Delete line 145/146 ‘all animals in….humanely’ as you already mention this in line 139.
We revised this text passage accordingly.
Line 147: subheader should be called ‘animals, husbandry and housing’ and should include animal information like line 164-170 data. In line 165, you write ‘randomly’. I would delete ‘randomly’.
We revised this text passage accordingly.
Line 148: ‘kept’ should be replaced by ‘housed’. Instead of 0.5m2 and 1m2, provide the width and depth of the cages.
We revised this text passage accordingly.
Line 158: ‘food regimen was slight changed’, I assume you meant that daily diet was variated by adding….right?
Yes, that is what we meant. We have added a corresponding passage.
Line 165, 180 and 184: ‘chosen’ should be replaced to ‘selected’.
We revised this text passage accordingly.
Line 179: ad ‘due arythmia’ already here.
We revised this text passage accordingly.
Line 181: ‘result‘ should be ‘resulted’
We revised this text passage accordingly.
Line 185: why did you use potassium chloride and not a barbiturate for euthanasia?
We use potassium chloride instead of an overdose of barbiturate to prevent crystalline deposits in the heart (cells), which would interfere with the subsequent histological processing and staining. We added this information accordingly to the manuscript (line 191-193)
Line 196: tail vein is named ‘caudal vein’. Latin names like vena saphena should be in latin (in the whole manuscript)
We revised this text passage and the whole manuscript accordingly.
Line 198-203: it is was it is, but why did propofol and remifentanil need additional isoflurane as anesthetic? The flow rate is missing. Same for line 253-256: why providing double anesthesia? If you need to ventilate them to control breathing movements, you can only do isoflurane/sevoflurane so no propofol needed, right?
We are aware about this might confusing setting. You are right most animals do not need additional inhalation anesthetics when being under the control of propofol and opioids (fentanyl, remifentanil). It was not possible to achieve sufficient depth of anesthesia in our animals using the TiVA (total intravenous anesthesia) without not causing significant volume-induced strain. Therefore, an inhalation anesthetic was added, also with the knowledge that this may have an influence on the PV-loop measurements. We included the flow rate of the isoflurane/sevoflurane in the manuscript.
208: ‘figure 6 j’ should be ‘figure 2 j’. Normally, figure names should occur in line of the text so J can not be named until A-I is mentioned.
We revised this text passage and the whole manuscript accordingly.
Figure 2: all images are so small that they are hard to interpret. As this is an online journal, present bigger images.
We rearranged the size of the images in Figure 2 and revised the figure in the manuscript accordingly.
Line 230: before hemodynamic measurements was called PV loop. Keep one line otherwise it is getting confusing.
We revised this text passage and the following sections accordingly.
It is a suggestion: to keep clearance for the readers, one could use subheaders like
2 Materials and methods
2.1 Animals, husbandry and housing
2.2 Study design
2.3 MRI
2.4 Echocardiography
2.5 PV loop
2.6 statistics
This because the results are also divided in subheaders 3.1 PV loop 3.2 MRI 3,.3 echocardiography and in addition, it reads easier when all information about a procedure is written in one subheader.
Thank you very much for this suggestion. We revised the text passage accordingly.
Line 243: ‘Figure 6 G’ should be ‘Figure 2 G’ and again, mention 2G before 2A-2F is not common.
We revised the whole manuscript accordingly.
Line 270: are this the same tubes in 247-248? If yes, then one can delete it in 270.
Yes, these are the same tubes and we revised the corresponding passage in the manuscript.
Line 280: again, Figure 6. Please change that in the whole manuscript
We revised the whole manuscript accordingly.
Line 340: SV, EF, Co, EDV…etc are first mentioned here. It would help to have them full written instead of only abbreviated and ended in the end in a table, as a reader will think he/she missed the first time the abbreviation was used so he/she will check all pages before this one to find the abbreviation instead looking in the end of the paper. In line 381 they are full written but that is after line 340. Please adapt.
We revised the corresponding passages accordingly.
Line 381: instead of shown one could use a more scientific word like presented.
We revised the manuscript accordingly.
Line 394: you write ‘see also Table 2’ but I suppose this should be Table 3. In Table ‘3’ the title is also mentioned above the caption, delete. Idem line 412. Table 3 or 2?
It is Table 3, we revised the whole manuscript accordingly.
In Line 412 I think Figure 2E and 2F are mentioned instead of Figure 2.
We revised the manuscript accordingly.
Figure 4 and 5 are presented before they are mentioned in the manuscript. Confusing!
We revised the manuscript accordingly. Now all figures are presented after they are mentioned in the manuscript.
Line 541: the method of choice could be ‘the gold standard’.
We revised the manuscript accordingly.
Line 614: “One important factor is also the choice of the anesthetic regimen” could be “One important factor is the anesthetic regimen: for……”
We revised the manuscript accordingly.
References: there are double references in the list e.g.
Number 10 = 49
NUMBER 13 =47
Number 20=42
Number 23=54
Number 76=50
Number 12=82
And I didn’t check them all…
Line 501 : reference 37 : this publication doesn’t discuss the availability of marmosets
Idem reference 38
We cross checked all references and revised the manuscript accordingly. Reference 37 and 38 are used according to their content in the supplements. (now reference 43 and 68 after the supplements)
Comments on the Quality of English Language
In the whole manuscript, more scientific words should be used like ‘our findings’ in line 29 and 34 could be ‘our results’. Same for line 568, 575, 592 and 754 where ‘found’ should be replaced by e.g., assessed, determined, observed, measured. ‘Chosen’ should be ‘selected’ in several lines e.g., line 454. Line 247 ‘directly after'=subsequently etc etc
We revised the whole manuscript accordingly.
Reviewer 2 Report
I have reviewed the manuscript entitled 'Functional cardiovascular characterization of the common marmoset (Callithrix jacchus)'
The manuscript appears to contribute to the literature a lot.
The role artificial intelligence systems are very important to direct current medicine. Heart diseases are the one of the most important to direct AI systems. With the aid of these models, we can also direct invasive approach in heart failure patients. Please add a short section to the discussion explaining the role of AI systems citing 'The Role of Artificial Intelligence in Coronary Artery Disease and Atrial Fibrillation'.
Author Response
Dear Reviewers,
Thank you very much for taking the time to critically review our manuscript. Your remarks and questions are highly appreciated; we did our best to address them in our revision. In the following lines, we gladly respond to your comments and elaborate on every point you kindly brought to our attention in our work:
- Reviewer 2:
The role artificial intelligence systems are very important to direct current medicine. Heart diseases are the one of the most important to direct AI systems. With the aid of these models, we can also direct invasive approach in heart failure patients. Please add a short section to the discussion explaining the role of AI systems citing 'The Role of Artificial Intelligence in Coronary Artery Disease and Atrial Fibrillation'.
Thank you for your comment with regard to this point of view. We have added a paragraph including a citation on AI in heart failure to fit the topic of this special issue as to your suggestion to the discussion.
Round 2
Reviewer 1 Report
Dear authors,
thank you for taking my comments into consideration and for implementing most.
Just some small details left:
-in the manuscript, at least 18x 'we' is used and 10x 'our'. Can you please rewrite those sentences? Just rephrase all those lines e.g., line 28 the obtained results were compared with general established methods......line 34: the results will provide the basis to establish....line 42. Therefore, a cardiac functional analysis was performed etc. Please rephase all we and our lines.
-in Line 74 you use the abbreviation NHP, please use that in line 84 and 501
-line 151: from 'quality....to waterworks' can be deleted. Ad libitum is latin so should be in italics
-line 177-180: some 'space bar' errors.
- Line 31, 276 and 547: chosen should be replaced by 'selected'
- Line 341: SVTau abbreviation is not explained.
- Line 568: should be 'to the best of the authors knowledge,...'
- ref 29: Copyright © 2023 looks weird. I assume a mistake.
In conclusion: nice masterpiece!
